# Psychological Vulnerability Indices and the Adolescent’s Good Mental Health Factors: A Correlational Study in a Sample of Portuguese Adolescents

**DOI:** 10.3390/children9121961

**Published:** 2022-12-14

**Authors:** Joana Nobre, Henrique Luis, Ana Paula Oliveira, Francisco Monteiro, Raul Cordeiro, Carlos Sequeira, Carme Ferré-Grau

**Affiliations:** 1Health School, Polytechnic Institute of Portalegre, 7300-555 Portalegre, Portugal; 2Faculty of Nursing, University of Rovira i Virgili, 43003 Tarragona, Spain; 3Nursing Research Unit for South and Islands (NURSE’IN), Polytechnic Institute of Setúbal, 2914-503 Setúbal, Portugal; 4VALORIZA—Research Centre for Endogenous Resource Valorization, Polytechnic Institute of Portalegre, 7300-555 Portalegre, Portugal; 5Unidade de Investigação em Ciências Orais e Biomédicas (UICOB), Faculdade de Medicina Dentária, Universidade de Lisboa, Rua Teresa Ambrósio, 1600-277 Lisbon, Portugal; 6Center for Innovative Care and Health Technology (ciTechcare), Polytechnic of Leiria, 2410-541 Leiria, Portugal; 7Comprehensive Health Research Centre (CHRC), University of Évora, 7000-811 Évora, Portugal; 8Group Inovation & Development in Nursing (NursID), Centro de Investigação em Tecnologias e Serviços de Saúde (CINTESIS), 4200-450 Porto, Portugal; 9Nursing School of Porto, 4200-072 Porto, Portugal

**Keywords:** adolescents, mental health, psychological vulnerability, schools

## Abstract

Background: Psychological vulnerability (PV) indicates the individual’s inability to adapt to stressful situations. Adolescents experience negative impacts on their future mental health if they do not acquire the skills and knowledge necessary to have good mental health during their developmental stage. Aim: To compare the PV index among the three stages of adolescence and to explore the factors involved in good mental health, including the relationship between adolescents’ PV indices and sociodemographic variables, and the relationship between adolescents’ PV index and their knowledge of the factors that characterize good mental health. Method: An exploratory, cross-sectional, correlational study was carried out in three public schools in a region of Portugal, using online self-completed questionnaires: the Psychological Vulnerability Scale (PVS) and the Mental Health-Promoting Knowledge (MHPK-10). Results: Our convenience sample consisted of 260 adolescents, with a mean age of 14.07 years who were students between 5th and 12th grades, mostly female. Moderate PV indexes were obtained that were higher in late adolescence, i.e., in older adolescents, who were females in a more advanced school year, with worse self-perceptions of their physical and mental health and body image, and who frequently used a health service due to mental health problems. The association between the PV index and the level of knowledge about the factors involved in good mental health did not reach a statistical significance (*p* = 0.06). Conclusions: These results suggest a need for a design of personalized interventions that promote adolescents’ mental health literacy, that prevent PV, and that should be initiated in early adolescence.

## 1. Introduction

Adolescents are a potentially vulnerable group in the community when it comes to mental health, mainly due to the profound and rapid changes that occur in this life cycle stage, both physically and mentally [1,2]. The rapid growth and development of the body and brain that occurs during this transitional phase between childhood and adulthood cause adolescents to undergo physical changes as well as changes in the way they think and solve problems, changes in the way they deal with emotions, and changes in social relationships [2,3]. These changes will have positive or negative impacts on adult life, depending on how effectively they are processed during adolescence, thus determining the evident vulnerability that adolescents are naturally prone to.

This vulnerability becomes visible when we look at the prevalence of mental health problems in around 14% of the global adolescent population [4]. It becomes worrisome when we see that the literature contends that more than 50% of cases of mental health problems in adults begin before the age of 14 [5].

According to the World Health Organization [6], adolescence is the period of an individual’s life cycle between the ages of 10 and 19. This period of life, characterized by profound biological, psychological/mental, and social transformations, is quite long; for this reason, several authors have divided it into stages. In our study, we adopted a classification system for adolescence consisting of three stages: early adolescence (10–14 years), middle adolescence (15–16 years) and late adolescence (17–19 years) [3,7,8].

Early adolescence is characterized by concrete thinking; the need for parental and peer approval; a strong identification and dependence on the best friend; the compelling influence of peer group standards on moral reasoning; therefore, there is a strong and widespread need for belonging [2,3,7].

Middle adolescence is characterized by the development of abstract thinking, although there is recourse to concrete thinking in stressful situations; the creation of body self-image; a great need to find his/her own skills and identity; more competitiveness with oneself and others; the beginnings of establishing more stable interpersonal relationships; thus, it is a stage marked by competence and uniqueness [3,7].

Late adolescence is a stage in which abstract thinking predominates; there is greater competence in problem solving; there is an intense search for the meaning of life on a personal level and for one’s own standards of morality and integrity; the adolescent displays the ability to make plans for the future and to set long-term goals; there is less interference from the opinions of others in one’s own decisions and choices; the adolescent feels comfortable with his/her body image; the individual demonstrates self-esteem and more skills in interpersonal relationships; thus, it is a stage in which worthiness is evident [2,3,7].

Conceptually, psychological vulnerability (PV) is defined as “a pattern of cognitive beliefs reflecting a dependence on achievement or external sources of affirmation for one’s sense of self-worth” [9] (p. 120), which translates into a tendency for the individual to have a negative perception of himself and the world where he interfaces with others, when faced with events considered stressful, reflecting the individual’s inability to adapt functionally to stress and becoming more fragile to it [9,10,11]. The literature in this context shows that perceived dependence, perfectionism, generalized negative attributions, and the need for external sources of approval are related to PV. Therefore, PV can be considered an indicator of individual’s deficit in coping behaviors [9,10,11].

In recent years, some studies have been published on PV showing that adolescents and young adults have moderate levels of PV [11,12,13,14], that there is a negative association between PV and adaptive constructs [15] and a positive relationship between PV and negative health outcomes [11,12], that female individuals have higher levels of PV [11,12,14], and that there is a negative relationship between resource literacy for good mental health and PV [16], all of which use the Psychological Vulnerability Scale [9], although it should be noted that most of these studies targeted higher education students.

Knowledge about the factors for obtaining and maintaining good mental health is one of the components of mental health literacy [17] and the one that has a salutogenic dimension; it is linked to mental health promotion. Regarding mental health literacy (MHL), in our study we adopted as a theoretical reference the concept of MHL defined by Jorm et al. [18] and its evolution over time [17,19], as well as the conceptual model of health literacy developed by Sørensen et al. [20], which involves “the knowledge, motivation and competencies of accessing, understanding, appraising and applying health-related information within the healthcare, disease prevention and health promotion setting, respectively” (p. 80). Research so far shows that adolescents have a good level of knowledge about the factors that promote good mental health [21], that girls have slightly higher levels of knowledge than boys, and that higher levels of knowledge of the factors that promote good mental health are associated with higher levels of well-being [22]. Some very recent studies in this area, but in the population of higher education students, show a good level of knowledge among the students [14], that female students and older students have higher levels of knowledge [14,23], and that a negative relationship exists between the PV index and the level of knowledge about the factors for obtaining and maintaining good mental health [16].

Given this global scenario, in an attempt to promote the mental well-being of all citizens and prevent the onset of mental health problems, in accordance with Goal 3, “Good Health and Well-Being” of the Sustainable Development Goals [24], the World Health Organization defends in the Comprehensive Mental Health Action Plan 2013–2030 [5] the need to continue to focus on strategies to promote mental health by implementing interventions in the early stages of the life cycle (childhood and adolescence) [5,25,26].

The relationship between the PV and knowledge about the factors for obtaining and maintaining good mental health was assessed in a sample of Portuguese university students by Teixeira et al. [16]. However, no studies were found in adolescents, so it is important to carry out this research because it offers the opportunity to promote future research studies that effectively personalize interventions to increase the adolescent’s mental health.

The primary objective of our study was to explore the relationship between adolescents’ PV index and their knowledge about the factors for good mental health, with the purpose of obtaining information to design an intervention to promote literacy on positive mental health among adolescents. Therefore, we outlined the following research questions:What is the PV index in early, middle, and late adolescence?What is the relationship between sociodemographic variables and adolescents’ PV?What is the level of knowledge about the factors for obtaining and maintaining good mental health in early, middle, and late adolescence?What is the relationship between PV and knowledge about the factors for obtaining and maintaining good mental health among adolescents?

## 2. Materials and Methods

### 2.1. Study Design

A descriptive, exploratory, cross-sectional, and correlational study was carried out, included in the quantitative research paradigm. The STROBE Statement checklist [27] was used as a guide for writing this article.

### 2.2. Setting

This study used, to collect the data, online questionnaires via Google^®^ Forms applied through a link that was emailed to participants from three public schools (5th to 12nd school year) belonging to a district in the Alentejo region of Portugal. Data collection was conducted between 17 April 2020 and 30 June 2020.

### 2.3. Participants

For the development of the study, we determined that we would need a sample made up of participants who cumulatively meet the following inclusion criteria: (a) being adolescents aged between 10 and 19 years old; (b) attend one of three public schools (5th to 12th grade) located in a district of the Alentejo region of Portugal, whose directors authorized the study; (c) have permission to participate in the study by their guardians/legal representatives provided through an informed consent form online; (d) agree to participate voluntarily in the study.

Participants were selected using a non-probabilistic and intentional sampling method. Thus, adolescents who met the inclusion criteria mentioned above and to whom the team of researchers had access constituted the convenience sample of this study.

We calculated the required size for our sample using the MGH Biostatistics Center Sample Size Calculator [28] for a margin of error of 5% and reliability of 90%, verifying that we would need at least 242 participants for our study. From a population of 2125 adolescents, we obtained 273 completed online questionnaires, of which 13 were eliminated because they were repeated, resulting in a total of 260 adolescents with questionnaires completely answered in our convenience sample.

To collect data from the participants, an email was sent to the school directors with a link to the informed consent form available on Google^®^ Forms to be sent to the parents/legal representatives. A document explaining the study in terms of methodology, possible risks/benefits, and ethical aspects was attached to this email. Subsequently, the link to the questionnaires to be completed by the adolescents on the Google^®^ Forms platform was sent to the email address that the parents/legal representatives had indicated when completing the informed consent forms. Participants were guaranteed the anonymity of the data collected.

### 2.4. Data Sources/Measurement

To provide answers to the research questions outlined, adolescents were evaluated in a single moment in relation to their sociodemographic characteristics, the value of their psychological vulnerability index, and their level of knowledge of factors that promote good mental health.

Sociodemographic characteristics were evaluated through a questionnaire constructed by the team of researchers, which was pre-tested. It is a questionnaire composed of 29 items that allowed the collection of the following data: age, gender, school year, employment and occupation situations of the father and mother; previous and current mental health problems, psychological monitoring, medication use; self-perception of mental health, physical health, and body image.

To assess the PV index of adolescents, we used **the Psychological Vulnerability Scale** [PVS] [11]. The PVS is an instrument originally developed by Sinclair & Wallston [9], which assesses PV by identifying inadequate cognitive patterns (perfectionism, dependence, need for external sources of approval, widespread negative attributions). The PVS has a one-dimensional structure and consists of 6 self-filling items, on a Likert scale of 1 to 5 points (1 = does not describe anything about me; 5 = describes me very well). The total score corresponds to the sum of all items and ranges between 6 and 39 points, where the highest scores correspond to the highest PV. The PVS was translated into Portuguese and validated for the Portuguese population by Nogueira et al. [11] and has good internal consistency, with a Cronbach α = 0.73. In the current study, the α Cronbach was 0.78.

We applied the questionnaire **the Mental Health-Promoting Knowledge** [MHPK-10] [29] to evaluate the level of knowledge about the factors that promote the attainment and maintenance of good mental health, in the dimensions of autonomy, relationship, and competence [22]. It was created by Bjørnsen et al. [21], and was translated into Portuguese and validated for the Portuguese population by Chaves et al. [29]. The MHPK-10 is a one-dimensional questionnaire composed of 10 self-filling items quoted on a Likert scale from 1 to 5 (1 = totally disagree, 5 = fully agree; option “N/A” = not applicable and equals zero points). In terms of score, the highest values correspond to a higher level of knowledge of the factors that promote the attainment and maintenance of good mental health, and a score with a mean of less than 4 reveals an insufficient level of knowledge, considering that the authors of the scale determined that the correct answers for each item would correspond to values 4 and 5 [21]. The MHPK-10 showed good internal validity, with a α Cronbach of 0.87 in the Portuguese version and 0.85 in our study.

### 2.5. Data Analysis

The SPSS^®^ version 27 (IBM Corp, Armonk, NY, USA) for Windows was used to perform statistical analysis. All results with *p* < 0.05 were considered statistically significant. We used descriptive statistics (absolute and relative frequencies, mean and standard deviation) to describe the variables according to their typology (qualitative/quantitative). We performed a prior test of the distribution of the variables and verified that our sample does not present normality (Shapiro Wilks test *p* < 0.001). Therefore, in terms of inferential statistics, we used non-parametric tests, specifically the Eta test to evaluate the association between nominal variables and PVS, and the Spearman correlation coefficient to verify the relationships between quantitative variables and PVS and between PVS and MHPK-10.

## 3. Results

### 3.1. Participant’s Characteristics

Our sample consisted of 260 adolescents, more girls than boys (55.8% versus 44.2%), and a mean age of 14.07 years (SD = 1.96), with a minimum age of 10 years and a maximum age of 19 years, the majority (59.6%) of whom attended 7th–9th grade, 27% attended 10th–12th grade, and 13.5% attended 5th–6th grade, and whose parents represented an active professional situation (father employed in 95.8% of cases and mother employed in 90.8% of cases). Most adolescents reported having good mental health, i.e., 98.5% did not have a diagnosed mental health problem; 69.6% reported not currently having psychological monitoring, 98.1% reported not taking medication on a regular basis for a mental disorder, and 97.7% reported not having used a health service in the past three months for changes in their mental health. The adolescents in our sample reported a mean score of 4.13 (SD = 0.82) related to the self-perception of their physical health and a mean score of 4.25 (SD = 0.89) of their mental health self-perception, both corresponding to a “Good” level. They also reported their body image self-perception as “Normal”, with a mean score of 3.73 (SD = 0.96).

### 3.2. Psychological Vulnerability Indices

The mean of the PV index (assessed by the PVS) of the adolescents in our sample was 14.71 (SD = 5.43), on a scale of 6 to 39 points, where 39 corresponds to the highest value of PV. The value found in this study was approximately in the middle of the scale, so we considered it to be at a moderate level (Table 1). The item “*6. I often feel resentful when others take advantage of me*” is the one in which adolescents have the highest vulnerability value (M = 3.34, SD = 1.44).

In a more detailed analysis, we found that it is in late adolescence that the PV index value is highest (M = 17.25, SD = 6.47), followed by middle adolescence (M = 15.09, SD = 5.24) and early adolescence (M = 13.99, SD = 5.14). Across all stages of adolescence in our sample, the item “*6. I often feel resentful when others take advantage of me*” has the highest values of the PV index, noting that in late adolescence also the item “*5. I tend to set goals that are too high and then feel frustrated trying to achieve them*” has high values of PV (Table 1).

To investigate the relationship between the sociodemographic variables and the PV indexes of adolescents, we used inferential statistical analysis. The Eta test showed a statistically significant association between PV and the use of a health service in the past three months for changes in the mental health of the adolescents (Eta = 0.046) although with a weak negative Spearman correlation (r_s_ (260) = −0.008, *p* = 0.902), indicating that adolescents who used a health service the least because of a mental health problem had a lower PV index. On the other hand, Spearman’s correlation coefficient revealed a moderately negative relationship between PV and mental health self-perception (r_s_ (260) = −0.355, *p* < 0.001) and body image self-perception (r_s_ (260) = −0.382, *p* < 0.001); a weak negative relationship between PV and physical health self-perception (r_s_ (260) = −0.289, *p* < 0.001); and a weak positive relationship between PV and age (r_s_ (260) = 0.195, *p* = 0.002) and school year (r_s_ (260) = 0.200, *p* = 0.001). We also calculated the mean of PV according to the variable sex, and the results showed that girls have a higher PV index (M = 15.65; SD = 5.59) than boys (M = 13.55; SD = 5.02). However, we did not find a statistically significant association between the variable sex and the PVS index. The analysis did not reveal other statistically significant relationships between the PV and the remaining sociodemographic variables.

### 3.3. Knowledge of the Factors for Good Mental Health

According to the analysis that Table 2 shows, we found that, overall, in our sample the adolescents reported a mean score of 4.49 (SD = 0.66) in MHPK-10, which reveals a sufficient level of knowledge of factors that promote good mental health, since the mean score is higher than 4. The MHPK-10 item in which adolescents had, in general, the lowest value was “*1. Handling stressful situations in a good manner*” (M = 4.31, SD = 1.17).

We also found that participants who are in middle adolescence reported the lowest mean score of knowledge about the factors that promote the attainment and maintenance of good mental health (M = 4.43, SD = 0.77), followed by those who are in early adolescence (M = 4.49, SD = 0.64) and then those in late adolescence (M = 4.55, SD = 0.51), although in all three stages of adolescence the mean is always higher than four. If we look at all items assessed by MHPK-10, we find that it is also in middle adolescence that the items have slightly lower values, except for items “*2. Believing in yourself*” and “*7. Mastering your own negative thoughts*” which show lower values in early adolescence, and items “*5. Setting limits for your own action*” and “*8. Setting limits for what is OK for me*” that have lower values in late adolescence.

### 3.4. Correlation between PVS and MHPK-10

To assess the relationship between PV and knowledge about the factors that promote good mental health we used the Spearman correlation coefficient. A statistically significant but weak negative relationship (r_s_ (260) = −0.181, *p* = 0.003) was found, meaning that a higher rate of PV is associated with a lower level of knowledge about the factors that promote good mental health.

A one-way Multivariate Analysis of Variance (MANOVA) was performed to examine whether differences existed in scores on the PVS and on the MHPK-10 between adolescents. Results of the evaluation of the assumptions of normality and homogeneity of variance-covariance matrices indicate that the homogeneity of covariance matrices across groups is not assumed (*p* < 0.01); linearity and multicollinearity were not satisfactory. With the use of Wilks’ criterion, results showed no statistical differences for sex (*p* = 0.564), age (*p* = 0.231), and mental health status (*p* = 0.591).

After a multivariable regression analysis to examine a relationship between PVS and knowledge when the other variables are statistically controlled, it is possible to see that, PVS accounts for 2% of the knowledge variable variance with statistical significance (*p* = 0.023). When the other variables are added to the model (age, sex, school year, employment and occupation situation of the father and mother; previous and current mental health problems, psychological monitoring, medication use; self-perception of mental health and physical health), PVS accounts for 10.5% of the knowledge variable. This second block of variables is responsible for 8.5% of the variance of the knowledge variable, but it is not statistically significant (*p* = 0.508). Individually we can see that when the other variables are added to the model, the PVS is not statistically significant anymore (*p* = 0.060), with the same happening to all the other variables.

## 4. Discussion

One of the research questions of our study sought to compare the PV index between the three stages of adolescence. The results demonstrate that in general the adolescents in our sample showed a moderate level of PV, which is in agreement with the results of the Alves study [13]. When comparing the three stages of adolescence, it was detected that in our sample PV is higher in late adolescence. We also found that, at this stage, adolescents reported feelings of disappointment, frustration, and helplessness in social relations (evidenced by the results of item 6 of the PVS), as well as a certain tendency towards perfectionism (evidenced by the results of item 5 of the PVS). Since we did not find any other studies in this field, we cannot compare the results. However, it should be noted that these negative feelings and expectations reported by adolescents can contribute to the development of feelings of hopelessness and failure in this target group [9], since it is at this stage that adolescents are developing the cognitive ability to make plans and outline goals for the future, to develop their morality and integrity, and the capacity for personal appreciation (self-esteem) [3,7], which can be compromised if adolescents do not handle their own expectations well, making them psychologically vulnerable [30] to the onset of mental health problems. These results make evident the need to invest in mental health promotion interventions in early adolescence, to reduce PV in late adolescence.

With the second research question we wanted to explore the relationship between sociodemographic variables and the PV index. The results showed that in our sample the adolescents who presented the highest levels of PV were those who reported worse self-perception of their mental health, body image and physical health, and were those who had attended a health service more frequently in the past three months for changes in their mental health, which indicates that PV is associated with negative health outcomes, fitting with the findings of Nogueira et al. [11], although in a sample of higher education students. In addition, older adolescents and those who were in a more advanced school year were those who also had higher levels of PV, suggesting signs of unadapted cognitive reactions to stressful events [9,11] that in late adolescence can be experienced in relation to future plans/expectations and to increasing pressure for academic, professional, and social success [2,3,7]. We also found that girls reported higher levels of PV than boys, which agrees with the findings from studies on PV in other populations [11,12,14,31,32]. However, in our sample of adolescents, this association was not statistically significant, in line with the studies by Nogueira [12], Nogueira et al. [11], and Nogueira et al. [31], but in disagreement with the studies by Sequeira et al. [14] and Yamaguchi et al. [32].

The third research question was intended to compare the level of knowledge about the factors for good mental health in the three stages of adolescence. The results of our study show a sufficient level of knowledge among the adolescents in our sample overall, which is in line with the results of the studies by Bjørnsen [21,22]. The totality of the participants in our sample revealed a low level of knowledge on how to adequately cope with stressful events, pointing to a need to improve skills in this context. When analyzing by adolescence stage, we found that participants in middle adolescence reported the lowest levels of knowledge, despite being at the sufficient level in this area, and participants in late adolescence were the ones who had the highest levels of knowledge about the factors for good mental health. Since there are no other studies so far that have done this comparative analysis, it is not possible to compare the results. Looking at these results in light of the specific aspects of adolescence highlighted in the literature, it makes perfect sense that in late adolescence, knowledge about the factors for good mental health is greater, due to the neuroplasticity of the adolescent’s brain [25], which allows for the development of abstract thinking at this stage.

The fourth research question was intended to investigate the relationship between adolescents’ PV index and their level of knowledge about the factors for obtaining and maintaining good mental health. The results show that the participants in our sample presented a negative—marginally and not statistically significant—relationship between PVS and knowledge; that is, adolescents who presented a higher PV index reported lower levels of knowledge about the factors that promote good mental health. This relationship had not yet been explored in adolescents, but the results of a study in this area on a sample of higher education students [16] were recently published, in which this negative relationship was also evidenced between PV and knowledge about factors promoting good mental health. However, it is important to mention that we have detected a certain incongruence in the results presented here by our study, because given this negative relationship and given that we previously identified that in late adolescence the level of knowledge about the factors promoting good mental health is higher, this should imply that in late adolescence the PV index should be lower, but that is not what we verified in the results we obtained. Therefore, this fact points to the need to implement interventions promoting mental health literacy [33,34,35], specifically at the level of knowledge utilization, because apparently adolescents have had access to information and have the knowledge, but lack the ability to apply it [20]. Such interventions should aim to prevent reactions revealing PV in the transitions that adolescents face, which may compromise future transitions in the life cycle [36].

When interpreting the results of our study, it should be considered that it has some limitations. One of these limitations is that a non-probabilistic sampling method was used with a convenience sample, making it impossible to generalize the results obtained. Another limitation concerns the type of study conducted, a cross-sectional study, because this type does not admit the establishment of causality between the variables under study. Another potential limitation is related to data collection that is based on adolescents’ self-reporting, so the risk of bias should be considered due to the possibility that the responses were given according to social desirability. Finally, it is important to take into consideration that the data collection took place during the beginning of the COVID-19 pandemic, more specifically during the first confinement in Portugal, so this factor may have influenced the adolescents’ answers and the results obtained.

Despite the limitations, it is also important to mention the strengths of this study. One strength is that the data collection instruments used have good psychometric properties, which provide robust data. Another strength is that the results highlight the importance of promoting the mental health of adolescents.

## 5. Conclusions

Our study results show that adolescents in our sample are moderately psychologically vulnerable, especially those in late adolescence. They also show that being older and in a more advanced school year, being a female, attending health services more frequently in relation to a mental health problem, having worse self-perceptions of body image and both physical and mental health, are all characteristics associated with higher rates of PV in our sample. Furthermore, the results of the present study indicate that adolescents in general report a good level of knowledge about the factors for obtaining and maintaining good mental health, with middle adolescence being the stage at which the level of knowledge is lowest and late adolescence being the stage at which it is highest. Finally, this study did not find a statistically significant relationship between PV and knowledge about the factors for good mental health. This clearly suggests the need to focus on individual and/or community personalization of early-stage adolescent mental health interventions, based on a prior assessment.

Future research is needed on adolescents in other geographical locations and larger samples, using longitudinal studies to see if these results continue over time, as well as experimental or quasi-experimental studies to test the effectiveness of interventions that promote the mental health of adolescents and, thus, prevent their PV.

This study has implications for clinical practice, since it provides a diagnosis of the situation of the PV index of adolescents, alerting health professionals, teachers of adolescents, and researchers to the need to implement interventions/programs that promote the mental health of adolescents and their mental health literacy—across all adolescents, especially the most psychologically vulnerable ones. These interventions/programs should start as early as possible to reduce psychological vulnerability throughout the various stages of adolescence and in the transition to adulthood.

## Figures and Tables

**Table 1 children-09-01961-t001:** PVS according to the adolescence stage (*n* = 260).

PVS Item	Total			Early Adolescence			Middle Adolescence			Late Adolescence		
	*n*	Mean	SD	*n*	Mean	SD	*n*	Mean	SD	*n*	Mean	SD
1. When I can’t achieve my goals, I feel like a failure as a person.	260	2.45	1.33	153	2.25	1.22	75	2.64	1.38	32	2.97	1.51
2. I feel I deserve better treatment than I normally get from others	260	2.41	1.28	153	2.25	1.26	75	2.55	1.27	32	2.84	1.32
3. I am well aware that I often feel inferior to others.	260	2.01	1.34	153	1.84	1.21	75	2.17	1.42	32	2.47	1.59
4. I need approval from others to feel good about myself.	260	1.90	1.17	153	1.75	1.08	75	2.03	1.23	32	2.31	1.35
5. I tend to set goals that are too high and then feel frustrated trying to achieve them.	260	2.60	1.33	153	2.54	1.31	75	2.49	1.30	32	3.09	1.42
6. I often feel resentful when others take advantage of me.	260	3.34	1.44	153	3.36	1.49	75	3.21	1.37	32	3.56	1.37
PVS Total Score	260	14.71	5.43	153	13.99	5.14	75	15.09	5.24	32	17.25	6.47

Abbreviations: *n*, number of cases; PVS, Psychological Vulnerability Scale; SD, standard deviation.

**Table 2 children-09-01961-t002:** MHPK-10 according to the adolescence stage (*n* = 260).

MHPK-10 Item	Total			Early Adolescence			Middle Adolescence			Late Adolescence		
	*n*	Mean	SD	*n*	Mean	SD	*n*	Mean	SD	*n*	Mean	SD
1. Handling stressful situations in a good manner	260	4.31	1.17	153	4.28	1.19	75	4.25	1.24	32	4.59	0.87
2. Believing in yourself	260	4.69	0.86	153	4.65	0.95	75	4.69	0.80	32	4.88	0.42
3. Having good sleep routines	260	4.65	0.91	153	4.67	0.87	75	4.59	1.05	32	4.69	0.78
4. Making decisions based on own will	260	4.35	0.95	153	4.46	0.83	75	4.17	1.10	32	4.22	1.04
5. Setting limits for your own action	260	4.41	1.07	153	4.45	1.00	75	4.36	1.13	32	4.31	1.26
6. Feeling that you belong in a community	260	4.41	1.14	153	4.44	1.10	75	4.27	1.32	32	4.59	0.84
7. Mastering your own negative thoughts	260	4.62	0.91	153	4.53	1.03	75	4.73	0.74	32	4.78	0.61
8. Setting limits for what is OK for me	260	4.49	1.06	153	4.45	1.08	75	4.60	0.94	32	4.41	1.19
9. Feeling valuable regardless of your own accomplishments	260	4.42	1.11	153	4.50	1.03	75	4.21	1.33	32	4.47	0.84
10. Experiencing school mastery	260	4.53	1.03	153	4.56	0.97	75	4.45	1.15	32	4.59	0.98
MHPK-10 Total Score	260	4.49	0.66	153	4.49	0.64	75	4.43	0.77	32	4.55	0.51

Abbreviations: *n*, number of cases; MHPK-10, Mental Health Positive Knowledge; SD, standard deviation.

## Data Availability

Data available on request due to ethical restrictions.

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
