# Peer review of "Psychological Vulnerability Indices and the Adolescent’s Good Mental Health Factors: A Correlational Study in a Sample of Portuguese Adolescents"

_children, 2022, doi:10.3390/children9121961_

Round 1
Reviewer 1 Report (Previous Reviewer 3)
Dear Authors,
I believe that the revised version of the manuscript meets all the conditions for publication. Congratulations for your work.
Minore:
- Please ensure that the abstract has the following elements: Background:1-2 sentences on the context and the need for the study; Method: 1-2 sentences on the methodology; Results: the majority of the abstract on the actual results of the study; Conclusions:1-2 sentences on key conclusions and recommendations
Author Response
Thank you for your interest in our article entitled “Psychological Vulnerability Indices and Adolescent’s Good Mental Health Factors: a correlational study in a sample of Portuguese adolescents”, manuscript ID: children-2080907.
We want to thank the Reviewer #1 for the constructive comments provided, which will surely result in a significant improvement of our paper.
Please find a revised version attached, which includes modifications according to the reviewers’ comments, and also other modifications we decided to do to improve the manuscript, all of them duly marked by the Track Changes function. We also address each of the comments below with our answers and hope that this work can now be accepted for publication.
Kind regards,
Joana Nobre
on behalf of all authors

Reviewer 2 Report (Previous Reviewer 1)
Comment
Regarding comment#9 which previously I gave a comment the authors (Sep. 2022):
The authors administered MANOVA to respond to the previous reviewers' comments. However I simply think that the authors should conduct a multivariable regression analysis to examine a independent relationship between PVS and knowledge, when the other variables (including sex, age, and mental health status) are statistically controlled.
Author Response
Thank you for your interest in our article entitled “Psychological Vulnerability Indices and Adolescent’s Good Mental Health Factors: a correlational study in a sample of Portuguese adolescents”, manuscript ID: children-2080907.
We want to thank the Reviewer #2 for the constructive comments provided, which will surely result in a significant improvement of our paper.
Please find a revised version attached, which includes modifications according to the reviewers’ comments, and also other modifications we decided to do to improve the manuscript, all of them duly marked by the Track Changes function. We also address each of the comments below with our answers and hope that this work can now be accepted for publication.
Kind regards,
Joana Nobre
on behalf of all authors

Round 2
Reviewer 2 Report (Previous Reviewer 1)
Comment
In the second round of the revised version of the manuscript, the authors conducted newly a multivariable regression analysis to examine the independent relationship between PVS and knowledge about promoting mental health. They reported a negative and marginally significant relationship between PVS and the knowledge (p=0.06). Since it was NOT significant (but marginally significant) in the case of the multivariable regression analysis, the revised version of the manuscript should be modified to fit the results, in the result of the abstract, and in the result, discussion, and conclusion section in the text body.
Author Response
Dear Reviewer #2,
Thank you for your interest in our article entitled “Psychological Vulnerability Indices and Adolescent’s Good Mental Health Factors: a correlational study in a sample of Portuguese adolescents”, manuscript ID: children-2080907.
We want to thank the Reviewer #2 for the constructive comment provided, which will certainly result in a significant improvement of our manuscript. Please find a revised version attached, which includes modifications according to the reviewer comment, and also other modifications we decided to do to improve the manuscript, all of them duly marked by the Track Changes function. We also address your comment below with our answers and hope that this manuscript can now be published.
COMMENT:
1.“In the second round of the revised version of the manuscript, the authors conducted newly a multivariable regression analysis to examine the independent relationship between PVS and knowledge about promoting mental health. They reported a negative and marginally significant relationship between PVS and the knowledge (p=0.06). Since it was NOT significant (but marginally significant) in the case of the multivariable regression analysis, the revised version of the manuscript should be modified to fit the results, in the result of the abstract, and in the result, discussion, and conclusion section in the text body.”
ANSWER: We thank to the Reviewer for this comment.
ACTIONS TAKEN:
The following modification has been added to the Abstract:
Lines 37-39:
There was aA negative and marginally significant relationship between the PV index and the level of knowledge about the factors for good mental health (p=0.06) was found. Conclusions: These results suggest the need for the design
The following modification has been added to the heading “4. Discussion”:
Lines 366-368:
The results show that the participants in our sample presented a negative, marginally significant relationship between PVS and the knowledge, that is, adolescents who presented a higher PV index, reported a lower level of knowledge about the factors that promote good mental health.
The following modification has been added to the heading “5. Conclusions”:
Lines 409-410:
Finally, this study found that higher rates of PV are related to lower levels of knowledge about the factors for good mental health in a marginally significant way.
Other improvements:
â–ª Some writing modifications were made in the Abstract:
Abstract: Background: Psychological vulnerability (PV) is an indicator ofindicates the individuals’ inability of individuals to adapt to everyday stressful situations. Adolescents may experience negative impacts on their future mental health if they do not acquire skills and knowledge on how to have good mental health during their developmental stage. Aim: To compare the PV index between among the three stages of adolescence as well as the factors for good mental health, exploring the relationship between the adolescents’ PV index and the sociodemographic variables, and also the relationship between the adolescents’ PV index and their knowledge about the factors for good mental health.
Method: An exploratory, cross-sectional, and correlational study was carried out in three public schools in a region of Portugal, using online self-completed questionnaires: the Psychological Vulnerability Scale (PVS) and The Mental Health-Promoting Knowledge (MHPK-10). Results: Our convenience sample
consisted of a total of 260 adolescents, with a mean age of about 14.07 years, studying between 5th and 12th grade, mostly female.
â–ª One more document was cited in the heading “1. Introduction”:
Line 119:
in the early stages of the life cycle (childhood and adolescence)
[5,23,]24].
[Naccarella, L.; Guo, S. A Health Equity Implementation Approach to Child Health Literacy Interventions. Children 2022, 9 (9), 1284. https://doi.org/10.3390/children9091284.]
â–ª Two more studies were cited in the heading “4. Discussion”:
Line 330:
which can be compromised if adolescents do not handle their own expectations well, making them psychologically vulnerable [29] to the onset of mental health problems.
[Demirci, İ.; EkÅŸi, H.; EkÅŸi, F.; Kaya, Ç. Character Strengths and Psychological Vulnerability: The Mediating Role of Resilience. Curr. Psychol. 2021, 40 (11), 5626–5636. https://doi.org/10.1007/s12144-019-00533-1.]
Line 379:
Therefore, this fact points to the need to implement interventions promoting mental health literacy[31,]32] specifically at the level of the knowledge utilization
[Baxter, A.; Wei, Y.; Kutcher, S.; Cawthorpe, D. School-Based Mental Health Literacy Training Shifts the Quantity and Quality of Referrals to Tertiary Child and Adolescent Mental Health Services: A Western Canada Regional Study. PLoS One 2022, 17 (11), e0277695. https://doi.org/10.1371/journal.pone.0277695.]

This manuscript is a resubmission of an earlier submission. The following is a list of the peer review reports and author responses from that submission.
Round 1
Reviewer 1 Report
Comment
This study examined the relationship between psychological vulnerability and mental health-promoting knowledge among adolescence in Portugal. This manuscript may contribute to this area of research. Further attention to the issues presented below would strengthen the manuscript.
#1
As the second paragraph of page 2 was too long, it could be divided into 2-3 paragraphs with lead sentences. I would like to suggest to use one paragraph for each early, middle, and late adolescence statement.
#2
Why is it capitalized? the term "Higher Education" in the last line of page 2.
#3
In the last part of the Introduction section, the authors again noted the research questions. I believe that it is unnecessary since it was already shown in the previous paragraph.
#4
Regarding the participants recruiting method. It states that the Google Form link was sent to the participants via email for data collection. It is necessary to describe in detail how this e-mail address or the list of participants was obtained. Although the authors stated that the study used "a non-probabilistic and intentional sampling method," without a specific description of the recruitment method, it is difficult to determine how representative the population used in this study is.
If contact information for all students in a particular school is available, this is a complete survey. Depending on the participation rate, if so, it can be considered highly representative.
For the reasons stated above, as the representativeness of the population cannot be determined, the validity of the results demonstrated for the mean of each indicator and the differences in them by age group cannot be determined.
#5
In the line 6 from the bottom of page 5, the authors stated that the value of PVS among the study participants was "at a moderate level". If there is any criteria for judging the level, it is necessary to clearly state it.
#6
I think tables 1 and 2 could be merged.
#7
I think table 3 should be deleted.
#8
I think tables 4 and 5 could be merged.
#9
In page 8, the authors reported a negative correlation between PVS and MHPK10 using a univariate analysis. However the authors should conduct to a multivariate analysis, controlling for basic characteristics of participants such as gender, age, and mental health status, I believe.
Reviewer 2 Report
The paper is very well structured, but the introduction could be improved by focusing more on the main question of the study.
Discussion can be more focused on the most important results and their practical implications.
Reviewer 3 Report
I am grateful to the editor for providing me with this opportunity to be among the first readers of this draft entitled “Psychological Vulnerability Indices and Adolescent’s Good Mental Health Factors: a correlational study”.
I have read the efforts of the authors to try to fill the gap in the academic research that takes care of mental health in adolescents. Nevertheless, some important issues should be definitely addressed before the paper is suitable for publication. I list them below.
Although I am not a native English speaker, I had some serious doubts regarding language correctness in some parts of the manuscript. I will leave it to the assessment of the paper’s editor, however I believe that the paper could benefit from a thorough proof-reading by a native English speaker.
- Title: please ad ’in Portuguese adolescent population’
- It’s not clear to me which is the primary endpoint of the study? What is the main hypothesis? The entire introduction section should be carefully re-read revised and shortened, in order to provide a clear description of the purpose of the work and its significance
- Considering the COVID-19 pandemic during data collection it is not clear to me why the authors did not include the influence of the pandemic on the design and the results of the study. In my opinion COVID-19 pandemic must be considered as a factor that might influence the participants responses and results
- It is not clear to me why authors have decided to adopt the three-stage classification of adolescence phase in their study? Please consider more recent psychological studies regarding adolescent development.
- Please add more literature in the introduction section.
- Consider adding more recent psychological theories to the description of the adolescent development stage. Better to cite the latest and most relevant studies from a similar topic.
- I am not sure what exactly did the authors want to say by stating: “In recent years, some studies have been published within the scope of PV that show that adolescents and young adults have moderate levels of PV (Line 94)” – according to which scale?
- Line 126-130 – the statement itself is not strong enough to justify the study – please add scientific background to define the purpose of the work and its significance
- Please provide references to the statement: “Although the inferential statistics did not reveal an association between PV and sex, we calculated the mean of PV according to sex, since other previous studies in other populations showed differences. (line 261-262)”
- Update the paper with recent references